# The Association of Gender and Mortality in Geriatric Trauma Patients

**DOI:** 10.3390/healthcare10081472

**Published:** 2022-08-05

**Authors:** Shreya Pandya, Timothy Le, Seleshi Demissie, Ahmed Zaky, Shadi Arjmand, Nikhil Patel, Lilamarie Moko, Juan Garces, Paula Rivera, Kiara Singer, Ivan Fedoriv, Zachery Garcia, James Kennedy, Bhavana Makkapati, Indraneil Mukherjee, Anita Szerszen, Jonathan Gross, Galina Glinik, Duraid Younan

**Affiliations:** 1The Department of Surgery, Division of Acute Care Surgery, Staten Island University Hospital, Staten Island, New York, NY 10305, USA; 2Biostatistics Unit, Feinstein Institutes for Medical Research, Staten Island University Hospital, Staten Island, New York, NY 10305, USA; 3Department of Anesthesia, The University of Alabama in Birmingham, Birmingham, AL 35294, USA; 4Department of Surgery, Division of Minimally Invasive Surgery, Staten Island University Hospital, Staten Island, New York, NY 10305, USA; 5Department of Medicine, Division of Geriatric Medicine, Staten Island University Hospital, Staten Island, New York, NY 10305, USA; 6Department of Orthopedics, Staten Island University Hospital, Staten Island, New York, NY 10305, USA

**Keywords:** gender, mortality, trauma

## Abstract

The association of gender with mortality in trauma remains a subject of debate. Geriatric trauma patients have a higher risk of mortality compared to younger patients. We sought to evaluate the association of gender with mortality in a group of geriatric trauma patients presenting to an academic level 1 trauma center (trauma center designated by New York State capable of handling the most severe injuries and most complex cases). Methods: We performed a retrospective review of geriatric trauma patients who were admitted to our trauma center between January 2018 and December 2020. Data collected included vital signs, demographics, injury, and clinical characteristics, laboratory data and outcome measures. The study controlled for co-morbidities, injury severity score (ISS), and systolic blood pressure (SBP) in the ED. Multivariable logistic regression analysis was performed to evaluate the association of gender and mortality. Results: 4432 geriatric patients were admitted during the study period, there were 1635 (36.9%) men and 3859 (87.2%) were White with an average age of 81 ± 8.5 years. The mean ISS was 6.7 ± 5.4 and average length of stay was 6 ± 6.3 days. There were 165 deaths. Male gender (OR 1.94, 95% CI 1.38 to 2.73), ISS (OR 1.12, 95% CI 1.09 to 1.14), Emergency Department SBP less than 90 mmHg (OR 6.17, 95% CI 3.17 to 12.01), and having more than one co-morbidity (OR 2.28, 95% CI 1.55 to 3.35) were independently predictive of death on multivariable logistic regression analysis. Conclusion: Male gender, Emergency Department systolic blood pressure less than 90 mmHg, having more than one co-morbidity, and injury severity are independent predictors of mortality among geriatric trauma patients.

## 1. Introduction

As the geriatric population continues to grow in the US (United States), there are increased efforts to develop healthcare standards optimal to their unique needs. It is estimated that by 2030, 1 in 5 Americans will be at least 65 years old. By 2060, the US Census projects an increase from 52 million (2018) to 95 million, with a rise from 16% to 23% of the population qualifying as geriatric [1]. One particular focus in healthcare that is expanding to meet the needs of the population is that of the patient sustaining traumatic injuries.

Although the typical management of the trauma patient is high acuity in nature, the geriatric trauma patient is unique, as they often have a greater number of preexisting comorbidities, polypharmacy, and subsequently experience a higher rate of complications and overall mortality [2]. Although the most common traumatic mechanism observed in the geriatric population is low-energy blunt trauma such as ground-level falls [3]; they are also as susceptible to other forms of trauma observed in the non-geriatric population. Unique to the geriatric population, is the discrepancy in survival between gender, as female geriatric patients have an overall higher rate of survival compared to male patient-sustained injuries of comparable severity [4].

The association of gender with mortality in trauma remains a subject of debate. Although there is substantial evidence for sex-related outcome differences [5], the underlying mechanism of gender-specific responses after acute injury remains to be established. Sex-related hormones are a possible explanation, as a high ratio of estrogen to androgen appears to be protective both in immunologic and inflammatory responses to traumatic injuries [5].

We sought to evaluate the association of gender with mortality in a group of geriatric trauma patients presenting to an academic trauma center between January 2018 and December 2020.

## 2. Materials and Methods

### 2.1. Procedures, Patient Selection, and Variable Definition

Patient consent as well as approval from both the Institutional Review Board (IRB) and ethics committee were obtained for our study. We performed a retrospective review of patients ≥ 65 years of age who arrived at Staten Island University Hospital as trauma activations between January 2018 and December 2020. This was a hospital designated by New York State as capable of handling the most severe injuries, also known as a Level 1 Trauma Center.

### 2.2. Measures, Inclusion Criteria, and Exclusion Criteria

All geriatric trauma patients who arrived as trauma activation over the study period were considered for inclusion. Data collected from the trauma database included demographics (age, gender, and race), mechanism of injury, injuries sustained, injury severity (injury severity score “ISS”), pre-existing co-morbidities (chronic renal failure “CRF”, diabetes mellitus “DM”, hypertension “HTN” and chronic obstructive pulmonary disease “COPD”); complications including deep vein thrombosis “DVT”, acute kidney injury “AKI”, ventilator-associated pneumonia “VAP”, pulmonary embolism “PE”, hospital length of stay (LOS), and survival. These measures were selected due to their correlation with clinical acuity and mortality. After the original analysis of the whole group of patients, they were divided into two groups based on gender for further analysis.

### 2.3. Statistical Methods

This is a retrospective cohort study, the cohorts being men and women. The primary comparison was between the two cohorts. We hypothesize that male gender is an independent predictor of death after adjusting for potential confounding factors. Categorical data were summarized by the number and percentage of patients falling within each category. Continuous variables were summarized by descriptive statistics including mean and standard deviation or median and interquartile range. The primary comparison was between the gender groups. The primary outcome variable is hospital mortality. Bivariate analyses were performed using the *χ*^2^-test, ANOVA, and Wilcoxon Rank-sum test, as appropriate. The independent effects of gender, hypotension in ED (SBP < 90), Injury Severity Score (ISS), and previous medical history (whether the patient has two or more comorbidities versus 1 or 0) were evaluated using a multivariable logistic regression analysis. The main variable of investigation was gender; hypotension (SBP < 90) upon presentation, Injury Severity Score (ISS) and having two or more comorbidities were chosen as they were important predictors of death in trauma [1,4,5,6,7,8,9,10,11,12,13,14]. All statistical tests and confidence intervals were two-sided. *p*-values < 0.05 were considered statistically significant. All statistical analyses were performed using SAS software (Statistical Analysis Systems Inc., Cary, NC, USA).

## 3. Results

A total of 4432 geriatric patients (age > 65 years) presented to our trauma center as trauma activation during the study period. A total of 1635 (36.9%) of the patients were Men, 3859 (87.2%) were White, and 20.5% had at least one comorbidity. The median injury severity score (ISS) was 5 (7) [15] and the median hospital length of stay (LOS) was 5 (5). Injury and clinical characteristics are demonstrated in Table 1; one hundred and sixty-five patients (3.72%) died.

When the patients were divided into two gender groups, the Men group had higher lactate (*p* < 0.01) and INR (*p* < 0.01) levels, a higher incidence of smoking (*p* = 0.03), and a higher incidence of head (*p* < 0.01), chest (*p* < 0.01), and abdomen (*p* < 0.01) injuries but a lower incidence of extremity injuries (*p* < 0.01); they also had a higher incidence of >1 pre-existing co-morbidity (*p* < 0.01) and higher death (*p* < 0.01) (Table 2).

The Male gender (OR 1.94 with 95% CI 1.38–2.73) compared to Female group, ED SBP < 90 (OR 6.17 with 95% CI 3.17–12.01), injury severity score (ISS) (OR 1.12, 95% CI 1.09 to 1.14) and the presence of more than one co-morbidity (OR 2.28, 95% CI 1.55 to 3.35) were independently predictive of death on the multi-variable logistic regression analysis (Table 3).

## 4. Discussion

We found that, among geriatric trauma patients, male gender is an independent predictor of death, controlling for shock at presentation to the Emergency Department, injury severity, and presence of co-morbidities. This finding has significant implications on the management of injury in this increasing patient population.

In a review of more than 18,000 blunt trauma patients admitted to a single major trauma center in 2001, Napolitano et al. did not find differences in mortality associated with gender among blunt trauma patients who did not develop pneumonia. However, it was found that male patients with pneumonia had increased mortality [13]. More recently, other authors have demonstrated differences in mortality based on gender associated with trauma and sepsis [2,9,12].

Gioffre et al.—while not comparing mortality—in a retrospective review of 4500 patients, did find that falls were more frequent in women and that disability had a bigger effect on elderly women than men [1]. Among elderly women, osteoporosis contributes significantly to the various extremity fractures. These injuries were demonstrated to have an increase in hospitalization, thus leading to further clinical and social problems. In our study, women sustained more extremity injuries and less head injury compared to men and had significantly lower mortality.

Choudhry et al., in a review of published papers on gender and trauma outcome, demonstrated that the depletion of male sex hormones blocked the suppression of immune function following trauma. In fact, the administration of estrogen in males prevented the suppression of the body’s immune function, thus suggesting that male sex hormones were suppressive to the body’s immune function. The results obtained from this study support the proposed thought that gender influences the immune response following injury [6]. While the role estrogen plays in the geriatric population is not clear and is theoretically diminished, our data support these findings.

The role the immune system plays in the response to trauma has been demonstrated. The differences noted in mortality after trauma among geriatric patients could be attributed to the role sex hormones play after injury, as has been demonstrated by multiple studies [6,9,14,16]. Such studies have demonstrated the protective role of estrogen with a contrasting immunosuppressive effect of androgens [14]. Despite this evidence, the heterogeneity of the hormonal status of the population at the time of injury is well-known, which makes the effects of these sex hormones variable. [8,9] This further complicates the interpretation of the differences noted in survival among geriatric trauma patients in this study and leaves room for more studies addressing the effects of hormones in this patient population.

The women in our study had a higher percentage of extremity injury but lower percentage of head, chest, and abdomen injuries. The median injury severity score (ISS) was a little higher than that for males (6 versus 5); we controlled for the effects of ISS, co-morbidities, and shock (SBP < 90) on presentation to the Emergency Department, and the difference in survival between men and women persisted on multi-variable logistic regression analysis. It is also to be noted that the women were older.

Using sex steroids as a therapeutic tool in the setting of trauma and sepsis to improve survival has been suggested due to the role that these hormones play in the response to injury [10,11,12]. However, these hormonal levels can differ in the population and might have even more variable levels in the geriatric population—when their levels should be much lower physiologically—when compared to younger patients, thus making the use of these hormones in the geriatric population questionable.

While the women in this study had lower INR and lactate upon presentation to the Emergency Department (ED), this does not fully explain their improved survival given that their injury severity score was higher. The women also had a lower proportion of patients with more than one comorbidity and a lower percentage had a smoking history; we accounted for the co-morbidities in the multi-variable logistic regression analysis where female gender was found to be independently associated with decreased mortality. Considering the findings in this article and the recent advances in genetic mapping, it is possible that patients’ management will also take gender into consideration.

Filipescu and Ştefan, in a review article, examined the differences between men, women, and transgender patients’ responses to anesthesia. They concluded that women are less sensitive to anesthetic drugs when compared to men and also recover faster from general anesthesia [17]. One noted difference was that women were more likely to have adverse cardiac events after surgery compared to men. These conclusions have also been illustrated in a review article by Westerman and Wegner who examined gender differences in regards to atrial fibrillation; they found that women were more likely to experience complications when receiving anti-arrhythmic drugs as well as catheter ablation for atrial fibrillation, and are at higher risk for stroke [18].

Our study has certain limitations. It is retrospective in nature, this limits the applicability of the findings. Most of the patients sustained low injury with a median injury severity score (ISS) of 5, and it is a single-center study. While we controlled for the co-morbidities these patients suffered form, we do not have the “cause of death” for these patients, which would have shed more light into the noted difference in outcome.

In conclusion, male gender in geriatric trauma patients is associated with death, independent of the severity of injury, comorbidities, and shock at presentation.

## Figures and Tables

**Table 1 healthcare-10-01472-t001:** Demographic, injury, clinical characteristics, and outcomes of geriatric trauma patients stratified by “Dead”, *n* = 4432.

	All	Dead	Alive	*p*
**Demographic**				
Sex, *n* (%)				
Male	1635 (36.9)	89	1546	<0.01
Race, *n* (%)				
White	3859 (87.2)	148	3711	0.01
Black	106 (2.4)	4	102	
Other	460 (10.4)	13	447	
Age (mean ± SD)	81 ± 9	84.5 ± 7	80.8 ± 9	<0.01
**Injury**				
ISS, median (IQR)	5 (7)	9 (12.5)	5 (7)	<0.01
Head, *n* (%)	1041 (23.5)	65	976	<0.01
Chest, *n* (%)	669 (15.09)	36	633	0.02
Abdomen, *n* (%)	146 (3.29)	10	136	0.07
Extremity, *n* (%)	2537 (57.24)	82	2455	0.05
**Clinical**				
ED SBP, mean ± SD	150 ± 32	138 ± 46	151 ± 31	<0.01
ED Heart rate, mean ± SD	81 ± 17	85 ± 26	81 ± 16	0.11
ED Respiratory rate, mean ± SD	19 ± 3	18 ± 7	19 ± 3	0.6
Hematocrit, mean ± SD	37.3 ± 8.8	36 ± 7	37 ± 9	0.02
INR, median (IQR)	1.1 (0.3)	1.2 (0.3)	1.1 (0.3)	<0.01
Lactate, median (IQR)	1.7 (1.2)	2.2 (3.3)	1.6 (1.2)	<0.01
>1 co-morbidity, *n* (%)	721 (16.27)	48	673	<0.01
Smoking, *n* (%)	244 (5.51)	10	234	0.42
VAP, *n* (%)	13 (0.29)	4	9	<0.01
PE, *n* (%)	5 (0.11)	1	4	0.17
DVT—deep venous thrombosis, *n* (%)	45 (1.0)	6	39	<0.01
AKI—acute kidney injury, *n* (%)	33 (0.74)	14	19	<0.01
**Outcome**				
Length of stay, median (IQR)	5 (5)	6 (8)	5 (5)	<0.01

ED = emergency department, SBP = Systolic blood pressure, INR = international normalization ratio, ISS = Injury Severity score, VAP = ventilator-associated pneumonia, and PE = pulmonary embolism.

**Table 2 healthcare-10-01472-t002:** Demographic, injury, clinical characteristics, and outcomes of geriatric trauma patients based on gender.

	Men *n* = 1635	Women *n* = 2797	*p*
**Demographic**			
Race, *n* (%)			
White	1398 (85.6)	2461 (88.1)	0.02
Black	51 (3.1)	55 (2.0)	
Other	184 (11.3)	276 (9.9)	
Age, Mean ± SD	79.3 ± 8.4	81.9 ± 8.5	<0.01
**Injury**			
Severity (ISS), median (IQR)	5 (7)	6 (5)	<0.01
Head, *n* (%)	459 (28.1)	582 (20.8)	<0.01
Chest, *n* (%)	287 (17.6)	382 (13.7)	<0.01
Abdomen, *n* (%)	71 (4.3)	75 (2.7)	<0.01
Extremity, *n* (%)	781 (47.8)	1756 (62.8)	<0.01
**Clinical**			
ED SBP, mean ± SD	146 ± 32	153 ± 41	<0.01
ED Heart rate, mean ± SD	81 ± 18	82 ± 16	0.10
ED Respiratory rate, mean ± SD	19 ± 3	19 ± 3	0.41
Hematocrit, mean ± SD	38.1 ± 12.3	36.7 ± 5.8	<0.01
INR, median (IQR)	1.1 (0.3)	1.1 (0.3)	<0.01
Lactate, median (IQR)	1.8 (1.3)	1.6 (1)	<0.01
>1 morbidity, *n* (%)	322 (19.7)	399 (14.3)	<0.01
Smoking, *n* (%)	99 (6.1)	145 (5.2)	0.03
VAP—ventilator-associated pneumonia, *n* (%)	7 (0.4)	6 (0.2)	0.10
PE—pulmonary embolism, *n* (%)	2 (0.1)	3 (0.1	0.34
DVT—deep venous thrombosis, *n* (%)	18 (1.1)	27 (0.9)	0.11
AKI—acute kidney injury, *n* (%)	13 (0.8)	20 (0.7)	0.14
**Outcome**			
LOS (days), median (IQR)	5 (6)	5 (4)	0.68
Died, *n* (%)	89 (5.4)	76 (2.7)	<0.01

ED = emergency department, SBP = Systolic blood pressure, and INR = international normalization ratio.

**Table 3 healthcare-10-01472-t003:** Multivariable logistic regression for factors predictive of death among geriatric trauma patients (*n* = 4432).

	OR	95% CI	*p*
Gender, male vs. female ED SBP < 90 vs. >90	1.94 6.17	1.38–2.73 3.17–12.01	<0.0001 <0.0001
ISS (injury severity score)	1.12	1.09–1.14	<0.0001
≥2 comorbidity compared to ≤1	2.28	1.55–3.35	<0.0001

ED = emergency department and SBP = Systolic blood pressure.

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
