# Peer review of "The Association of Gender and Mortality in Geriatric Trauma Patients"

_healthcare, 2022, doi:10.3390/healthcare10081472_

Round 1
Reviewer 1 Report
I would like to thank the authors for their revision. This is a technically well-constructed paper, and with a recent focus on the geriatric trauma patient in their literature including under triage (a euphemism for ignoring the frail, vulnerable elderly patient) I think these risk factors can certainly be used to ‘trigger’ emergency room doctors into thinking about whether these patients need trauma team activation (depending on protocol) or general medical physician review.
I have no problem with the message of the paper. I only have one question: are ISS and LOS truly parametric data sets? They are usually heavily skewed, and for LOS the SD crosses into negative territory which is usually a sign of skewing, for what is a positive only value (depending on concepts of space and time!). I see INR and Lactate are non-parametric. If they aren’t normally distributed, please change it to median (IQR).
Author Response
We thank the reviewer for the comment, it has been revised in the final manuscript. The edits are highlighted in the attachment.

Reviewer 2 Report
No comments.
Author Response
We thank the reviewer for their time and feedback
This manuscript is a resubmission of an earlier submission. The following is a list of the peer review reports and author responses from that submission.
Round 1
Reviewer 1 Report
Thank you for allowing me to review this manuscript. This manuscript entitled "The Association of Gender and Mortality in Geriatric Patients with Trauma 2".
It is an interesting and highly relevant article today, although it has several limitations that make it suitable for publication in this journal. These limitations are detailed below:
- In the introduction, it would be important to clearly indicate the importance and timeliness of the topic of study. There are also errors in the citations: The first citation indicated is a 9, instead of a 1. Review and adjust the regulations of the journal.
- The material and methods section does not reflect different important aspects such as the study design. Nor does it make reference to important ethical considerations. It would be necessary to indicate if the study has the authorization of the ethics committee.
- The results are presented in a clear and orderly manner. Also, an interpretation of them is reflected. However, in the tables the acronyms used in them are not always specified at the foot of the table.
- Regarding the discussion. The results are discussed in an orderly manner in the manuscript. However, more citations should be included, to offer more rigor. However, there are more limitations than those reflected in this section. Check. In line 156 there is a change of letter
- On the other hand, we more accurately recommend their involvement in clinical practice. Also, it would be interesting to state the lines of the future that the authors consider.
- Finally, in relation to the bibliographical references, the article shows an insufficient number of consulted articles, which supposes an important weakness of the manuscript.
nice job
Author Response
Reviewer One:
1. In the introduction, it would be important to clearly indicate the importance and timeliness of the topic of study. There are also errors in the citations: The first citation indicated is a 9, instead of a 1. Review and adjust the regulations of the journal.
Response: We would like to thank the reviewer for the comment, it has been revised.
2. The material and methods section does not reflect different important aspects such as the study design. Nor does it make reference to important ethical considerations. It would be necessary to indicate if the study has the authorization of the ethics committee.
Response: We would like to thank the reviewer for the comment. Yes, the study was approved by the institution review board (IRB) of the hospital, we added a statement to show that in “Methods” section.
3. The results are presented in a clear and orderly manner. Also, an interpretation of them is reflected. However, in the tables the acronyms used in them are not always specified at the foot of the table.
Response: We would like to thank the reviewer for the comment, we revised it.
Regarding the discussion. The results are discussed in an orderly manner in the manuscript. However, more citations should be included, to offer more rigor. However, there are more limitations than those reflected in this section. Check. In line 156 there is a change of letter.
Response: We would like to thank the reviewer for the comment, we revised it.
On the other hand, we more accurately recommend their involvement in clinical practice. Also, it would be interesting to state the lines of the future that the authors consider.
Response: We would like to thank the reviewer for the comment
Finally, in relation to the bibliographical references, the article shows an insufficient number of consulted articles, which supposes an important weakness of the manuscript. nice job
Response: We would like to thank the reviewer for the comment. We added more citations.
Reviewer 2 Report
The authors presented a multivariable logistic regression analysis on the mortality using the data collected in their trauma centers. I have a few questions regarding the statistical analysis and results reporting. I will re-review this paper after those comments are addressed.
- Please indicate the affiliation of the authors.
- In line 56, please elaborate on "our trauma centers" as it wasn't previously defined.
- If the outcome variable is mortality, please stratify Table 1 by death by adding two more columns for patients who are alive or dead, separately, with p-values similar to Table 2.
- In line 95-101 and Table 2, if the p value is less than 0.01, please write "p<0.01" instead of "p<0.00".
- In Table 2, are head, chest, abdomen, and extremity the four levels of the same categorical variable of injury? If so, the chi-squared test should be used for all levels together instead of performing the test for each level.
- The reported data type is incomplete shown in the first column of Table 2. For example, ED SBP, is mean + std used?
- In Table 3, why were only four covariates used in the multivariable logistic regression analysis?
- In Table 4, why ED SBP was dichotomized at 90?
Author Response
1. Please indicate the affiliation of the authors.
Response: We would like to thank the reviewer for the comment. It has been revised
2. In line 56, please elaborate on "our trauma centers" as it wasn't previously defined.
Response: We would like to thank the reviewer for the comment. It has been revised
3. If the outcome variable is mortality, please stratify Table 1 by death by adding two more columns for patients who are alive or dead, separately, with p-values similar to Table 2.
Response: We would like to thank the reviewer for the comment. The table has been stratified by death.
4. In line 95-101 and Table 2, if the p value is less than 0.01, please write "p<0.01" instead of "p<0.00".
Response: We thank the reviewer for the comment. This was revised.
5. In Table 2, are head, chest, abdomen, and extremity the four levels of the same categorical variable of injury? If so, the chi-squared test should be used for all levels together instead of performing the test for each level.
Response: We thank the reviewer for the comment. No, they are not four levels of injury. A patient can have one or more of these injuries and other injuries
6. The reported data type is incomplete shown in the first column of Table 2. For example, ED SBP, is mean + std used?
Response: We thank the reviewer for the comment. Age, SBP and HR – mean (SD). For the other continuous variables – median (IQR)
7. In Table 3, why were only four covariates used in the multivariable logistic regression analysis?
Response: We thank the reviewer for the comment. This is based on clinical considerations. These four covariates are believed to be the only clinically meaningful covariates. If there are suggestions to include other covariates, we can redo the analysis.
8. In Table 4, why ED SBP was dichotomized at 90?
Response: We thank the reviewer for the interesting comment. We chose the variables to include in the multi-variable analysis carefully. In addition to the main variable (gender), we chose hypotension (systolic blood pressure less than 90mmmHg) upon presentation, injury severity score (ISS) as they are important predictors of death in trauma. We also included 2 or more co-morbidities in the analysis as the group of patients in the study are geriatric and they can have differing morbidities and we wanted to control for that effect too.
Reviewer 3 Report
I would like to thank the authors for allowing me to review their article looking at independent variables associated with mortality. Data was extracted from a database. A multivariate regression model has been used. With large numbers, it is no suprise that nearly all variables are significant. The end model demonstrates that male sex, hypotension, severity of injury and comorbidities are associated with death.
The manuscript is well written, and all the formatting is appropriate. There are some inconsistencies with font (line 156) from previous revision/ editing. This is a 'classical' database study, with all the limitations of any retrospective look at prospectively collected data, ie. inability to correct for unknown confounders. These are well acknowledged in the discussion.
My only question is the selection for the logistic regression model. Was this on a priori basis or was there some sort of selection criteria from univariate analysis? I assume the former which is well justified, but maybe discuss it more in methods. And reference the variables to previous papers that justify their incorporation, and of course sex which is your hypothesis. Speaking of , put in a quick hypothesis for scientific purposes.
Author Response
1. I would like to thank the authors for allowing me to review their article looking at independent variables associated with mortality. Data was extracted from a database. A multivariate regression model has been used. With large numbers, it is no Suprise that nearly all variables are significant. The end model demonstrates that male sex, hypotension, severity of injury and comorbidities are associated with death. The manuscript is well written, and all the formatting is appropriate. There are some inconsistencies with font (line 156) from previous revision/ editing.
Response: We would like to thank the reviewer for the comment. This was revised.
2. This is a 'classical' database study, with all the limitations of any retrospective look at prospectively collected data, ie. inability to correct for unknown confounders. These are well acknowledged in the discussion. My only question is the selection for the logistic regression model. Was this on a priori basis or was there some sort of selection criteria from univariate analysis? I assume the former is well justified, but maybe discuss it more in methods.
Response: We thank the reviewer for the comment. The covariates were selected on a priori basis using clinical judgement
3. And reference the variables to previous papers that justify their incorporation, and of course sex which is your hypothesis. Speaking of, put in a quick hypothesis for scientific purposes.
Response: We would like to thank the reviewer for the comment. This was revised.
Round 2
Reviewer 2 Report
Thank you for addressing my comments. Here are some follow up questions.
1. If the multivariate analyses were performed based on clinical important variables, please add this to the method section.
Author Response
If the multivariate analyses were performed based on clinical important variables, please add this to the method section?
Response: We thank the reviewer for the comment. It has been added.
The independent effects of gender, hypotension in ED (SBP<90), Injury Severity Score (ISS) and previous medical history (whether patient has two or more comorbidities versus 1 or 0) were evaluated using a multivariable logistic regression analysis. The main variable of investigation was gender; hypotension (SBP <90) upon presentation, Injury Severity Score (ISS) and having two or more comorbidities were chosen as they were important predictors of death in trauma (1, 2, 3, 4, 5, 6, 7, 8, 12, 13, 14, 15).
